# Overexpression of the Rieske FeS protein of the Cytochrome $b_6f$ complex increases $C_4$ photosynthesis in *Setaria viridis*

Maria Ermakova [1], Patricia E. Lopez-Calcagno [2], Christine A. Raines[2], Robert T. Furbank[1] & Susanne von Caemmerer [1]

$C_4$ photosynthesis is characterised by a $CO_2$ concentrating mechanism that operates between mesophyll and bundle sheath cells increasing $CO_2$ partial pressure at the site of Rubisco and photosynthetic efficiency. Electron transport chains in both cell types supply ATP and NADPH for $C_4$ photosynthesis. Cytochrome $b_6f$ is a key control point of electron transport in $C_3$ plants. To study whether $C_4$ photosynthesis is limited by electron transport we constitutively overexpressed the Rieske FeS subunit in *Setaria viridis*. This resulted in a higher Cytochrome $b_6f$ content in mesophyll and bundle sheath cells without marked changes in the abundances of other photosynthetic proteins. Rieske overexpression plants showed better light conversion efficiency in both Photosystems and could generate higher proton-motive force across the thylakoid membrane underpinning an increase in $CO_2$ assimilation rate at ambient and saturating $CO_2$ and high light. Our results demonstrate that removing electron transport limitations can increase $C_4$ photosynthesis.

[1] Australian Research Council Centre of Excellence for Translational Photosynthesis, Division of Plant Science, Research School of Biology, The Australian National University, Acton, Australian Capital Territory 2601, Australia. [2] School of Biological Sciences, University of Essex, Wivenhoe Park, Colchester CO4 3SQ, UK. Correspondence and requests for materials should be addressed to M.E. (email: maria.ermakova@anu.edu.au)

Crop yield gains achieved during the Green Revolution by conventional plant breeding were not based on photosynthetic traits[1] and the theoretical maximum yield of the light conversion in crops is yet to be achieved[2]. Increasing photosynthetic efficiency by targeted genetic manipulation could potentially double the yield of crop plants[3]. Electron transfer reactions of photosynthesis (also called light reactions) supply ATP and NADPH essential for $CO_2$ assimilation and therefore are a target for improvement[4]. It has been demonstrated in $C_3$ plants that facilitating electron transport by overexpressing the components of electron transfer chain can result in higher assimilation rates[5,6]. As $C_4$ plants play a key role in world agriculture with maize and sorghum being major contributors to world food production and sugarcane, miscanthus and switchgrass being major plant sources of bioenergy[7], improvement of electron transport reactions could further increase the rates of $C_4$ photosynthesis and yield[8,9].

In $C_3$ plants the electron transport chain is localised to the thylakoid membranes of mesophyll cells and consists of four major protein complexes: Photosystem II (PSII), Cytochrome $b_6f$ (cyt$b_6f$), Photosystem I (PSI) and ATP synthase. The first three complexes sustain linear electron flow from the water-oxidising complex of PSII to NADP$^+$, the terminal acceptor of PSI, which is accompanied by acidification of the internal compartments of thylakoids (lumen). Transmembrane difference in electrochemical potentials of protons, or proton-motive force (pmf), serves as the driving force for ATP synthesis.

Cyt$b_6f$ oxidises plastoquinol reduced by PSII and reduces plastocyanin, which then diffuses to PSI; plastoquinol oxidation is a rate-limiting step in the intersystem chain[10]. As a result of Q-cycle operating between the two binding sites of cyt$b_6f$, two protons are translocated from the stroma to the lumen per one electron[11]. Once pools of inorganic phosphate are exhausted, ATP synthesis slows down and causes a build-up of the pmf across the membrane[12]. Both components of pmf, the pH gradient ($\Delta$pH) and the electrochemical gradient ($\Delta\Psi$), are equally capable of driving ATP synthesis[13]. However, the pH component has a major regulatory effect on the electron transport chain and at pH < 6 causes slowing down of the plastoquinol oxidation[14] and down-regulation of PSII activity via non-photochemical quenching (NPQ). NPQ is a mechanism that triggers the attenuation of PSII activity by the dissipation of excess light energy in the form of heat in the light-harvesting antenna of PSII (LHCII)[15]. Fast, pH-dependent NPQ component in higher plants, q$_E$, is regulated by the PHOTOSYSTEM II SUBUNIT S (PsbS) protein[16] and xanthophyll cycle[17].

Cyt$b_6f$ forms a homodimer where each monomer consists of eight subunits: major subunits, Rieske FeS protein (PetC), Cytochrome $b_6$ (PetB), Cytochrome f (PetA) and subunit IV (PetD), and minor subunits, PetG, PetL, PetM and PetN[18]. There is an increasing amount of evidence that the amount of Rieske FeS protein, one of the two nuclear-encoded subunits along with PetM, regulates the abundance of cyt$b_6f$[6,19–24]. Transgenic Arabidopsis thaliana plants overexpressing Rieske FeS showed an increase in the amounts of other cyt$b_6f$ subunits, positive effects on PSII electron transport rate and $CO_2$ assimilation rate and decreased NPQ[6]. Studies on transgenic tobacco plants indicate that cyt$b_6f$ determines the rate of electron transport through the electron transport chain and concomitantly the $CO_2$ assimilation rate[19–23].

$C_4$ photosynthesis is a biochemical $CO_2$ concentrating pathway operating between mesophyll (M) and bundle sheath (BS) cells and there are three biochemical $C_4$ subtypes[25]. PEP carboxylase (PEPC) catalyses primary carbon fixation in the cytoplasm of mesophyll cells into $C_4$ acids. In $C_4$ plants like maize, sorghum and setaria malate diffuses to the BS cells where it is

decarboxylated inside chloroplasts by NADP-malic enzyme (NADP-ME) to provide $CO_2$ for ribulose bisphosphate carboxylase oxygenase (Rubisco). Pyruvate resulting from malate decarboxylation diffuses back to mesophyll cells where it is regenerated into PEP by pyruvate ortophosphate dikinase. $C_4$ species with NADP-ME biochemistry require a minimum of 1 NADPH and 2 ATP in mesophyll cells and 1 NADPH and 3 ATP in BS cells per one $CO_2$ fixed[8]. The components of mesophyll electron transport chain are very similar to those described above for $C_3$ plants but BS cells of NADP-ME $C_4$ species are effectively supplied with NADPH via malate coming from the mesophyll cells and therefore are more specialised for ATP production. BS cells of NADP-ME species usually have little or no PSII activity and operate active cyclic electron flow (CEF) leading to the formation of pmf but not to NADP$^+$ reduction[26]. There are two pathways for CEF: one via PROTON GRADIENT REGULATION 5 protein (PGR5), cyt$b_6f$ and PSI, and another one via chloroplastic NAD(P)H:Quinone oxidoreductase 1-like complex (NDH complex), cyt$b_6f$ and PSI[27].

Since cyt$b_6f$ is a component of electron transport chain in both mesophyll and BS cells, we used Setaria viridis, a model NADP-ME-type $C_4$ plant, to study effects of constitutive Rieske FeS overexpression on $C_4$ photosynthesis. We demonstrate that in both cell types increased abundance of Rieske FeS results in higher content of cyt$b_6f$ and allows higher photosynthesis rates without notable changes of Rubisco and chlorophyll content. Our results indicate that in $C_4$ plants electron transport is one of the limitations for $CO_2$ assimilation, particularly at high light and non-limiting $CO_2$ concentrations, and it is under cyt$b_6f$ control.

## Results

**Generation of transgenic plants with Rieske overexpression.** For Rieske FeS overexpression, the coding sequence of PetC gene from Brachypodium distachyon (BdPetC) was codon-optimised for the Golden Gate cloning system and assembled into two constructs (230 and 231) under the control of the maize ubiquitin promoter (Supplementary Fig. 1). The constructs were transformed into S. viridis using stable agrobacterium-mediated transformation. Eleven $T_0$ transgenic plants were selected on the basis of hygromicin resistance and a subset of nine lines was analysed for Rieske FeS protein abundance, the presence of the BdPetC transcript and insertion numbers (Fig. 1a). Plants that went through the transformation process but tested negative for the insertion ("escapes") were used as control for $T_0$ plants and $T_1$ progeny. Rieske FeS and Rubisco large subunit protein abundances per leaf area were highly variable between transgenic and control plants in the $T_0$ generation (Fig. 1a), because $T_0$ plants have been regenerated from the tissue culture at different times and thus varied substantially in age. Therefore three lines, 230(4), 231(3) and 231(6), were selected for further analysis of the $T_1$ progeny based on the presence of the BdPetC transcript (Fig. 1a).

In the $T_1$ generation, a quantitative estimate of the changes in Rieske FeS levels from the immunoblots showed that a number of transgenic plants contained increased amounts of Rieske FeS protein per leaf area compared to control plants with a maximum increase of 10–15% in 230(4)-7, 231(3)-3 and 231(6)-1 plants (Fig. 1b, c). Seeds from those plants were collected and used for the analysis of their $T_2$ progeny. Interestingly, analysis of the BdPetC expression in the $T_1$ generation of the line 230(4) (Supplementary Fig. 2) demonstrated that, although homozygous plants had about twofold higher transcript abundance compared to heterozygous plants, only heterozygous plants showed an increase of Rieske FeS protein level (Fig. 1b). Similar effect was observed in $T_1$ plants of the line 231(3) and consequently also in the $T_2$ progenies of the lines 230(4)-7 and 231(3)-3: only some

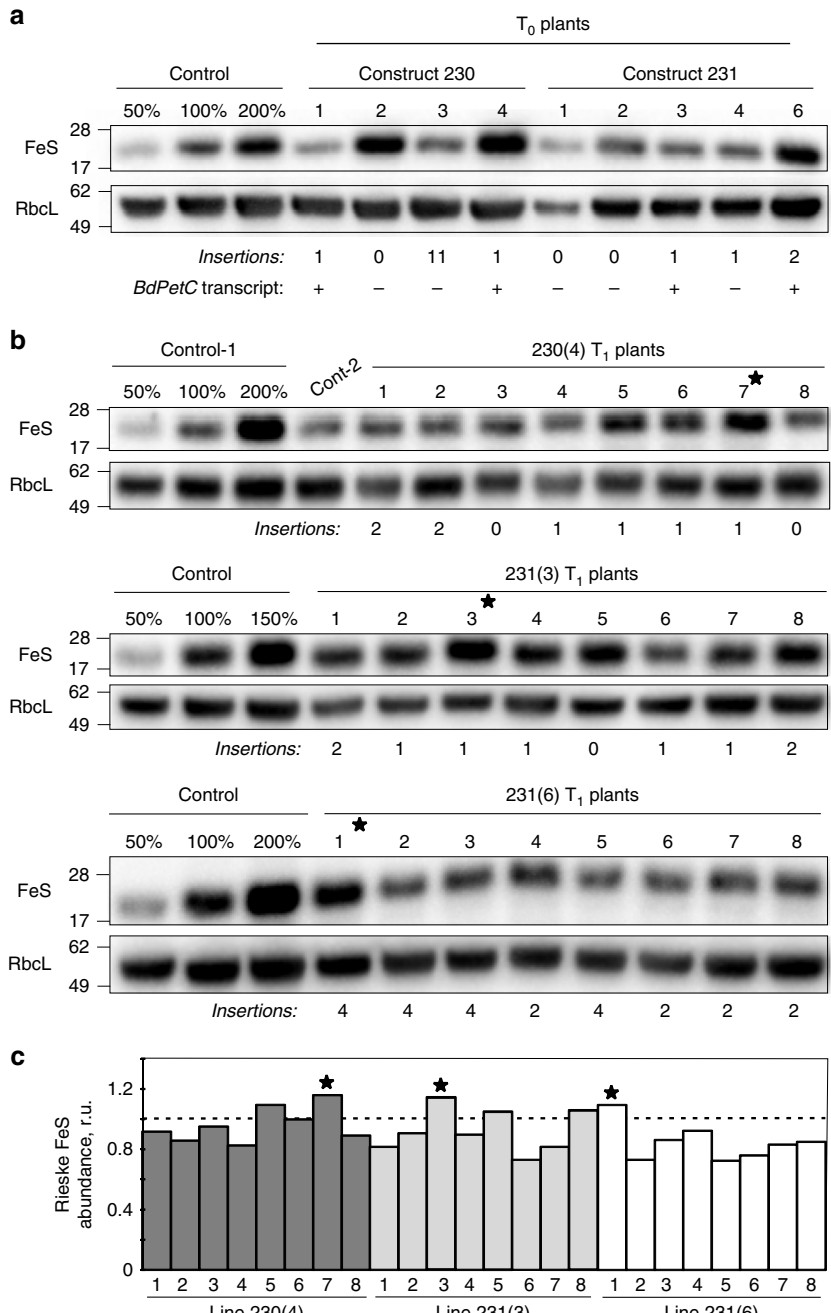

**Fig. 1** Selection of the transgenic *S. viridis* plants with Rieske FeS overexpression by immunodetection of Rieske FeS and Rubisco large subunit (RbcL) protein abundances on leaf area basis. **a** $T_0$ lines selected based on hygromycin resistance after transformation with constructs 230 and 231; insertion numbers indicate copy numbers of the hygromycin phosphotransferase gene; (+), transcript of the *PetC* from *B. distachyon* (*BdPetC*) was detected; (−) *BdPetC* transcript was not detected. **b** $T_1$ plants of three independent lines and their insertion numbers. **c**. Quantitative estimate of Rieske FeS abundance in $T_1$ plants from the immunoblots shown in (**b**) relative to control plants (=1). The plants that went through the transformation process and tested negative for the hygromycin phosphotransferase gene ("escapes") were used as control plants in the $T_0$ and $T_1$ generations. Positions of the molecular weight markers (kDa) are indicated on the left. Asterisks indicate plants selected for further analysis in the $T_2$ generation. Uncropped images of the membranes used for immunodetection are available in Supplementary Fig. 5. Data points for the item c are available in Supplementary data 1

heterozygous plants had increased Rieske FeS protein abundance (Supplementary Fig. 3). Moreover, Rieske FeS protein over-expression could only be detected when plants were grown at a low irradiance of 200 µmol m$^{-2}$ s$^{-1}$.

Transgenic plants with higher Rieske FeS protein abundance (FeS-OE hereafter) could be identified using pulse modulated chlorophyll fluorescence measurements of photochemical and non-photochemical processes at the incident light intensity with

the MultispeQ (Table 1). In $T_2$ progeny, FeS-OE plants, when compared to Zero(Null)-segregates control plants that are genetically identical to wild type, demonstrated significantly higher quantum yield of PSII (φPSII), a parameter estimating a proportion of the light absorbed by chlorophyll associated with PSII that is used in photochemistry, and a lower quantum yield of NPQ (φNPQ), i.e. a proportion of absorbed light actively dissipated in the PSII antennae. Electron transport parameters

**Table 1 Photosynthetic electron fluxes and leaf parameters in transgenic and control plants**

| Parameter | Control | Control-level FeS | FeS-OE |
|---|---|---|---|
| Relative chlorophyll (SPAD) | 50.79 ± 1.60 | 51.56 ± 1.61 | 51.78 ± 2.06 |
| Leaf thickness, mm | 0.76 ± 0.17 | 0.74 ± 0.12 | 0.72 ± 0.11 |
| φPSII | 0.598 ± 0.010 | 0.597 ± 0.004 | 0.624 ± 0.004* |
| φNPQ | 0.205 ± 0.011 | 0.200 ± 0.005 | 0.182 ± 0.003* |
| φNO | 0.197 ± 0.002 | 0.201 ± 0.001 | 0.194 ± 0.002 |
| Chlorophyll ($a+b$), g m$^{-2}$ | 0.39 ± 0.06 | | 0.38 ± 0.05 |
| Chlorophyll $a/b$ | 5.05 ± 0.13 | | 5.10 ± 0.10 |
| Rubisco, g m$^{-2}$ | 0.35 ± 0.03 | | 0.34 ± 0.02 |

"Control-level FeS", $T_2$ plants of the line 230(4)-7 with the control plants-levels of Rieske FeS protein; "FeS-OE", $T_2$ plants of the line 230(4)-7 with higher abundance of Rieske FeS per leaf area. MultispeQ measurements were performed at the incident irradiance of 200 μmol m$^{-2}$ s$^{-1}$, mean ± SE, $n = 16$ (4 biological replicates): φPSII quantum yield of PSII, φNPQ quantum yield of non-photochemical quenching, φNO quantum yield of non-regulatory energy dissipation, SPAD Soil Plant Analysis Development chlorophyll meter. Chlorophyll and Rubisco measurements are mean ± SE, $n = 3$ biological replicates. Asterisks indicate statistically significant differences between transgenic and control plants ($P < 0.05$). Zero(Null)-segregates were used as control plants

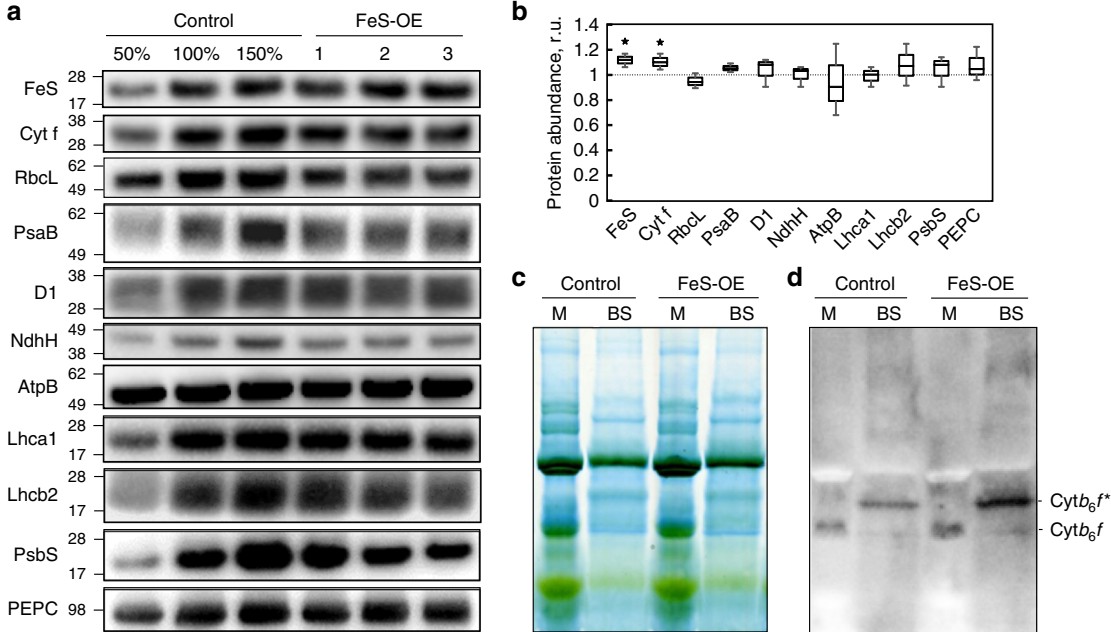

**Fig. 2** Relative abundance of photosynthetic proteins and Cytochrome $b_6 f$ complex in plants with Rieske FeS overexpression (FeS-OE) and control plants. **a** Western blots with antibodies against Rieske FeS, Cyt f (Cytochrome f), RbcL (Rubisco large subunit), PsaB (Photosystem I), D1 (Photosystem II), NdhH (NDH complex), AtpB (ATP synthase), Lhca1 (light-harvesting complex of Photosystem I), Lhcb2 (light-harvesting complex of Photosystem II), PsbS (Photosystem II subunit S) and PEPC (PEP carboxylase). Positions of the molecular weight markers (kDa) are indicated on the left. **b** Quantification of protein abundances in FeS-OE plants on leaf area basis relative to control plants (=1). Mean ± SE, $n = 3$ biological replicates. Asterisks indicate statistically significant differences between transgenic and control plants ($P < 0.05$). **c** Blue Native gel electrophoresis of the thylakoid protein complexes isolated from mesophyll (M) and bundle sheath (BS) cells; 10 μg of chlorophyll ($a+b$) loaded in each lane. **d** Immunodetection of Cytochrome $b_6 f$ complex from the Blue-Native gel with Rieske FeS antibodies; Cyt$b_6 f$ and Cyt$b_6 f$*—two distinct forms of the complex detected. Zero(Null)-segregates were used as control plants. Uncropped images of the membranes used for immunodetection are available in Supplementary Fig. 6. Data points for the item b are available in Supplementary data 1

measured in transgenic plants harbouring the transgene but showing a control plant-level of Rieske FeS ("Control-level FeS" in Table 1) did not differ from control plants. Other parameters such as relative chlorophyll content, measured by Soil Plant Analysis Development (SPAD) chlorophyll metre, and leaf thickness did not differ between the three groups (Table 1).

**Rieske FeS overexpression increases Cytochrome $b_6 f$ content.** Protein abundance of Rieske FeS and Cytochrome f subunits of cyt$b_6 f$ was analysed on leaf area basis in the $T_2$ progeny of the line 230(4)-7 using specific antibodies (Fig. 2a). Both cyt$b_6 f$ subunits were significantly more abundant in FeS-OE plants relative to control plants (Fig. 2b). Immunoblotting with specific antibodies

against representative subunits of other thylakoid complexes suggested that there were no significant differences in abundances of PSI, PSII, ATP synthase, NDH complex and light-harvesting complexes of PSI and PSII between FeS-OE and control plants on the leaf level (Fig. 2a, b). In addition, no changes in PsbS abundance were detected. Abundances of Rubisco large subunit (RbcL) and PEPC in FeS-OE plants were also similar to the levels detected in control plants (Fig. 2a, b). In line with the immunoblotting results, amounts of Rubisco active centers in FeS-OE and control plants, estimated by [14C] carboxyarabinitol bisphosphate binding assay, did not differ and neither did leaf chlorophyll ($a + b$) content and chlorophyll $a/b$ ratio (Table 1).

To see if the higher level of Rieske FeS protein resulted in increased abundance of the whole $cytb_6f$ complex, thylakoid protein complexes from the BS and mesophyll cells of FeS-OE and control plants were separated using Blue Native gel electrophoresis. No major changes were found between transgenic and control plants in composition of complexes within the thylakoid membranes of each cell type (Fig. 2c). Thylakoid protein complexes were probed with Rieske FeS antibodies to obtain the relative abundance of $cytb_6f$. Whilst one band matching the $cytb_6f$ band reported previously[28] was detected in the mesophyll thylakoids, in the BS thylakoids two bands were recognised by the Rieske FeS antibodies: a minor one with a similar molecular weight to the mesophyll $cytb_6f$ and a major one with a slightly higher molecular weight ($cytb_6f$*) (Fig. 2d). Relative abundance of $cytb_6f$ was higher in the mesophyll thylakoids and $cytb_6f$* was more abundant in the BS thylakoids of FeS-OE plants compared to control plants (Fig. 2d).

**Rieske FeS overexpression increases $CO_2$ assimilation rate**. $CO_2$ assimilation rate was measured in the $T_1$ plants of the lines 230(4), 231(3) and 231(6) and the $T_2$ progenies of the lines 230(4)-7, 231(3)-3 and 231(6)-1 at 1500 $\mu$mol m$^{-2}$ s$^{-1}$ and $CO_2$ of 400 ppm and plotted against the relative abundance of Rieske FeS protein on leaf area basis estimated from the immunoblots shown on Fig. 1b, Fig. 2a and Supplementary Fig. 3. All three lines showed a positive correlation between Rieske FeS abundance and $CO_2$ assimilation rate, suggesting that higher abundance of $cytb_6f$ supported higher rates of $C_4$ photosynthesis (Fig. 3).

To study the effect of Rieske FeS overexpression on $CO_2$ assimilation rate in more detail, the $CO_2$ response of assimilation was examined at a constant irradiance of 1500 $\mu$mol m$^{-2}$ s$^{-1}$ in the $T_2$ progenies of the lines 230(4)-7 and 231(3)-3. At lower intercellular $CO_2$ partial pressure no differences in assimilation rates were found between two FeS-OE lines and control plants (Fig. 4a, Supplementary Fig. 4). However, at $C_i$ above 150 $\mu$bar FeS-OE plants had higher $CO_2$ assimilation rates with a significant increase at the $C_i$ above 250 $\mu$bar. Maximum $CO_2$ assimilation rates reached by FeS-OE plants of the line 230(4)-7 and control plants at high $C_i$ were 41.30 ± 0.89 and 38.14 ± 0.81 $\mu$mol m$^{-2}$ s$^{-1}$ (mean ± SE, $n = 3$, $P = 0.049$), respectively, indicating about 8% increase of photosynthesis in FeS-OE plants (Fig. 4a). FeS-OE plants of the line 231(3)-3 also demonstrated about 10% increase of $CO_2$ assimilation rates at ambient and high

$CO_2$ (Supplementary Fig. 4). $CO_2$ response of $\varphi$PSII measured in FeS-OE and control plants showed similar trends: no difference at low $C_i$ but significantly higher $\varphi$PSII in FeS-OE plants at $C_i$ above 250 $\mu$bar (Fig. 4b). Stomatal conductance in FeS-OE plants estimated from the gas-exchange measurements was higher at $C_i$ above 250 $\mu$bar but not significantly ($0.05 < P < 0.15$) (Fig. 4c).

The light response of $CO_2$ assimilation was examined at a constant $CO_2$ of 400 ppm. Assimilation rates did not differ between FeS-OE plants of the line 230(4)-7 and control plants at the range of irradiances below 1500 $\mu$mol m$^{-2}$ s$^{-1}$ (Fig. 4d). However at light levels above 1500 $\mu$mol m$^{-2}$ s$^{-1}$, FeS-OE plants had significantly higher assimilation rates compared to control plants with a maximum increase of about 8%. $\varphi$PSII measured concurrently was significantly higher in FeS-OE plants compared to control plants over the range of irradiances above 200 $\mu$mol m$^{-2}$ s$^{-1}$ (Fig. 4e) whilst stomatal conductance did not differ in transgenic plants independently of irradiance (Fig. 4f).

**Rieske FeS overexpression plants have lower NPQ**. NPQ as well as photochemical and non-photochemical yields of PSI at different irradiances were analysed in FeS-OE plants of the line 230(4)-7 and control plants with the Dual-PAM-100. FeS-OE plants had significantly lower NPQ at the irradiances above 220 $\mu$mol m$^{-2}$ s$^{-1}$ (Fig. 5a). The effective quantum yield of PSI ($\varphi$PSI) in FeS-OE plants was significantly higher compared to control plants at the irradiances above 57 $\mu$mol m$^{-2}$ s$^{-1}$ (Fig. 5b). $\varphi$ND, a non-photochemical loss due to the oxidised primary donor of PSI, was significantly lower in FeS-OE plants compared to control plants at the range of irradiances between 57 and 220 $\mu$mol m$^{-2}$ s$^{-1}$ (Fig. 5c). $\varphi$NA, a non-photochemical loss due to the reduced PSI acceptor, did not differ significantly between FeS-OE and control plants but FeS-OE showed a tendency to lower $\varphi$NA at high irradiances ($0.1 > P > 0.05$) (Fig. 5d).

Dark-interval relaxation of the electrochromic shift signal has been recorded to analyse the generation of $pmf$ in leaves at various irradiances. Total $pmf$ was significantly higher in FeS-OE plants at 1287 $\mu$mol m$^{-2}$ s$^{-1}$ (Fig. 5e). Decomposition of $pmf$ into electrochemical gradient ($\Delta\Psi$) and proton gradient ($\Delta$pH) components revealed that the higher $pmf$ at 1287 $\mu$mol m$^{-2}$ s$^{-1}$ was due to the higher $\Delta$pH (Fig. 5f, g). Fitting of the first-order relaxation kinetics of electrochromic shift signal showed that there was no difference in thylakoid proton conductivity ($g_H^+$) between FeS-OE and control plants irrespective of irradiance (Fig. 5h).

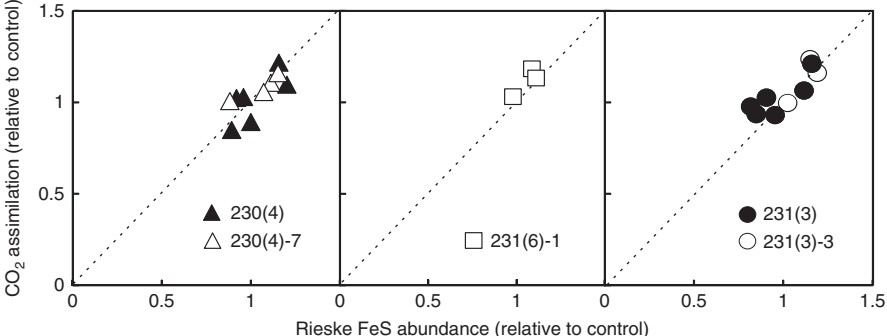

**Fig. 3** Relationships of Rieske FeS abundance and $CO_2$ assimilation rate in $T_1$ (filled symbols) and $T_2$ (open symbols) plants of three independent lines relative to control plants. Rieske FeS abundance was quantified from the immunoblots shown on the Fig. 1b, Fig. 2a and Supplementary Fig. 3. Steady-state $CO_2$ assimilation rates were measured at 1500 $\mu$mol m$^{-2}$ s$^{-1}$ and $CO_2$ partial pressure of 400 ppm. The plants that went through the transformation process and tested negative for the hygromycin phosphotransferase gene were used as control for $T_1$ plants and Zero(Null)-segregates were used as control for $T_2$ plants. Average rates of $CO_2$ assimilation measured from control plants were 26.47 $\mu$mol m$^{-2}$ s$^{-1}$ for $T_1$ plants and 32.65 $\mu$mol m$^{-2}$ s$^{-1}$ for $T_2$ plants. Uncropped images of the membranes used for immunodetection are available in Supplementary Fig. 5, Supplementary Fig. 6 and Supplementary Fig. 7. Data points for this item are available in Supplementary data 2

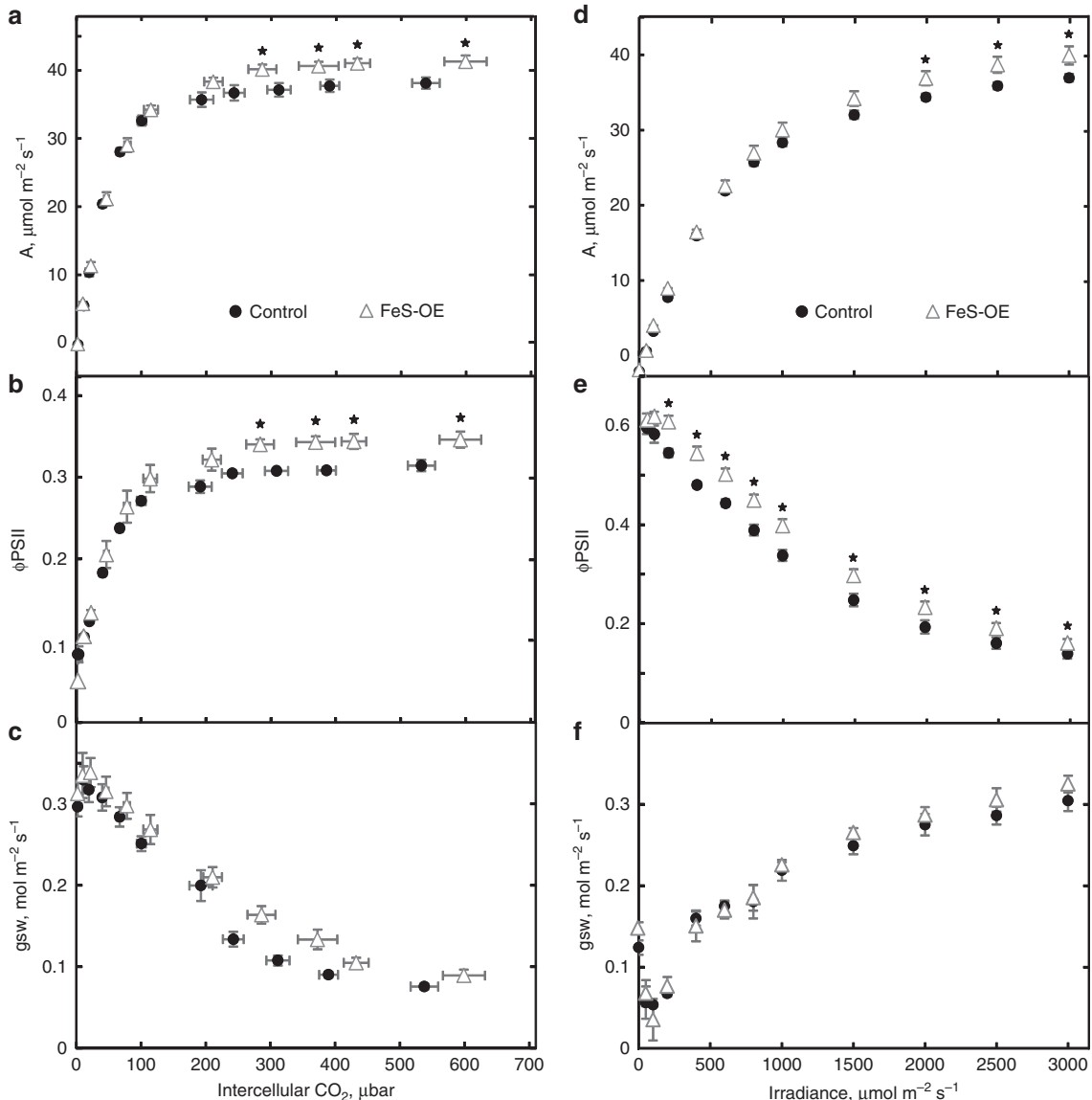

**Fig. 4** Gas-exchange and fluorescence analysis of plants with Rieske FeS overexpression (FeS-OE) and control plants. **a–c** $CO_2$ response of assimilation rate (A), quantum yield of Photosystem II ($\varphi$PSII) and stomatal conductance (gsw) at 1500 μmol m$^{-2}$ s$^{-1}$. **d–f** Light response of A, $\varphi$PSII and gsw at $CO_2$ partial pressure of 400 ppm. Mean ± SE, $n = 3$ biological replicates. Asterisks indicate statistically significant differences between FeS-OE and control plants ($P <$ 0.05). Results are for the $T_2$ progeny of the line 230(4)-7, results for the $T_2$ progeny of the line 231(3)-3 are shown on Supplementary Fig. 4. Zero(Null)-segregates were used as control plants. Data points are available in Supplementary data 3

Light-induced changes of photosynthetic parameters were studied on dark-adapted leaves over the course of illumination with red actinic light of 220 μmol m$^{-2}$ s$^{-1}$ (Fig. 6). During the first three minutes of illumination, FeS-OE plants had significantly lower NPQ indicating a slower induction of q$_E$ compared to control plants whilst no difference in NPQ relaxation kinetics was observed in darkness. Lower NPQ in FeS-OE plants was accompanied by the higher $\varphi$PSII during the first minutes of irradiance whilst $\varphi$PSI was higher in FeS-OE plants over the whole course of illumination compared to control plants (Fig. 6).

## Discussion

$C_4$ plants already have high photosynthetic rates when compared to $C_3$ plants which is achieved by operating a biochemical carbon concentrating mechanism, thereby reducing photorespiratory energy losses and allowing Rubisco to operate close to its

maximum capacity. However, further enhancing $C_4$ photosynthesis could have the potential for major agricultural impact. Maize, sorghum and sugarcane are among the most productive crops and all three belong to the NADP-ME biochemical subtype. Here, we used the closely related model NADP-ME species *S. viridis* to study the effect of increased electron transport capacity on $C_4$ photosynthesis. Since cyt$b_6f$ controls electron transport in $C_3$ plants and its abundance is likely determined by the abundance of Rieske FeS subunit[6,19], we attempted to accelerate electron transport rate in *S. viridis* by overexpressing Rieske FeS. As shown by immunodetection of the thylakoid protein complexes isolated from mesophyll and BS cells, constitutive overexpression of Rieske FeS protein in *S. viridis* resulted in higher abundance of the complete cyt$b_6f$ complex in both cell types (Fig. 2d).

The role of cyt$b_6f$ in the regulation of electron transport is defined by its position between the two photosystems. PSII

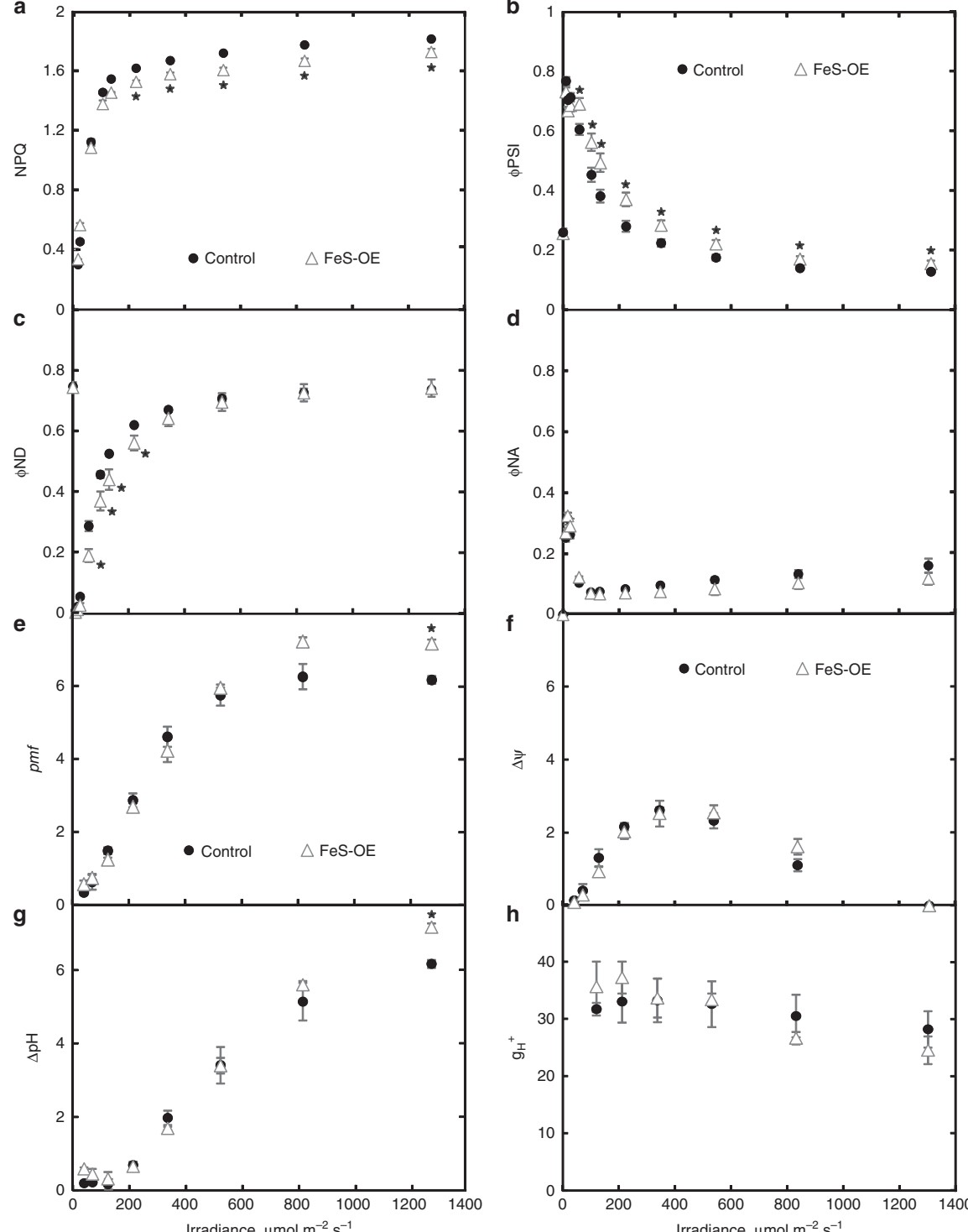

**Fig. 5** Light response of photosynthetic parameters in plants with Rieske FeS overexpression (FeS-OE) and control plants. **a** NPQ, non-photochemical quenching. **b** φPSI, quantum yield of PSI. **c** φND, non-photochemical loss due to the oxidised primary donor of PSI. **d** φNA, non-photochemical loss due to the reduced PSI acceptor. **e** *pmf*, proton-motive force, estimated from the electrochromic shift signal. **f** $\Delta\psi$, electrochemical gradient component of *pmf*. **g** $\Delta$pH, proton gradient component of *pmf*. **h** $g_H^+$, proton conductivity of the thylakoid membrane. Results are for the $T_2$ progeny of the line 230(4)-7. Mean ± SE, *n* = 3 biological replicates. Asterisks indicate statistically significant differences between FeS-OE and control plants (*P* < 0.05). Zero(Null)-segregates were used as control plants. Data points are available in Supplementary data 4

activity can be controlled by $cytb_6f$ levels either by limiting the rate of plastoquinol oxidation and hence oxidation of the primary electron acceptor $Q_A$ or by down-regulation of photochemical efficiency via NPQ. Plants generated here with more $cytb_6f$ had higher effective quantum yields of PSII at irradiances above

200 μmol m$^{-2}$ s$^{-1}$ which indicated that a higher portion of absorbed light reached PSII reaction centres (Fig. 4e). Lower NPQ measured in plants with higher $cytb_6f$ abundance at those irradiances suggested that the increase in φPSII was attributed to the lower loss of energy via thermal dissipation (Fig. 5a). Both

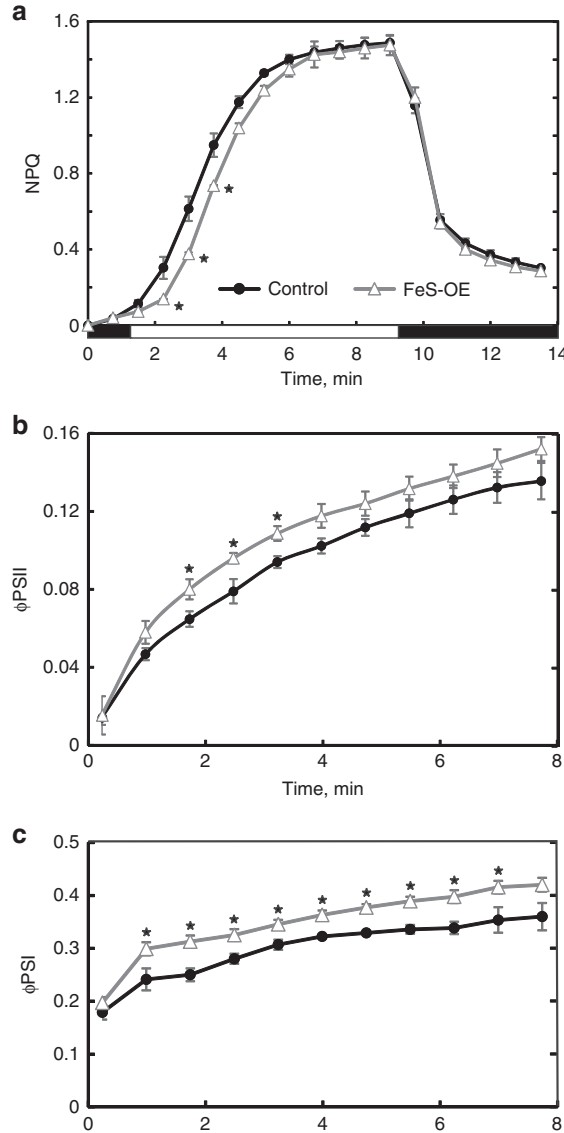

**Fig. 6** Photosynthetic parameters of plants with Rieske FeS overexpression (FeS-OE) and control plants during dark/light transitions measured on dark-adapted leaves. **a** Induction and relaxation of NPQ during dark-light-dark shift: black bars, darkness; white bar, red actinic light of 220 µmol m$^{-2}$ s$^{-1}$. **b**, **c** Dynamic changes of φPSII and φPSI, quantum yields of Photosystem II and Photosystem I, during first minutes of illumination with red actinic light of 220 µmol m$^{-2}$ s$^{-1}$. Results are for the T$_2$ plants of the line 230(4)-7. Mean ± SE, $n = 3$ biological replicates. Asterisks indicate statistically significant differences between FeS-OE and control plants ($P < 0.05$). Zero(Null)-segregates were used as control plants. Data points are available in Supplementary data 5

observations suggested that in FeS-OE plants Q$_A$ was more oxidised and linear electron flow was less limited by the rate of plastoquinol oxidation. Since in NADP-ME monocots mesophyll cells contain a much higher portion of PSII than BS cells (95–98%[29]), leaf level changes in PSII light-use efficiency can be attributed predominantly to mesophyll cells suggesting a higher capacity for whole chain electron transport in mesophyll cells. Steady-state electron flow measurements in leaves under the growth irradiance also confirmed the increased capacity for the linear electron flow in FeS-OE plants (Table 1).

PSI does not have an effective repair mechanism, such as the D1 protein turnover process of PSII[30], and when the transfer of electrons to PSI exceeds the capacity of the stromal acceptors, PSI photoinhibition causes long-term inhibitory effects on electron transport and carbon fixation[31,32]. Cyt$b_6f$ determines the amount of electrons reaching PSI from the intersystem chain, especially at low irradiance when CEF is mostly inactive[33]. In line with that, at irradiances below 340 µmol m$^{-2}$ s$^{-1}$ higher abundance of cyt$b_6f$ increased electron supply to the donor side of PSI (Fig. 5c) which resulted in the higher effective quantum yield of PSI (Fig. 5b).

Calculations of electron transport rates through photosystems are complicated in C$_4$ plants because of more complex distribution of light energy between PSI and PSII in the two-cell system. However, we attribute more efficient use of absorbed light for photochemical reactions in both photosystems to a "release" of cyt$b_6f$ control in both mesophyll and BS cells. In C$_3$ A. thaliana plants overexpressing Rieske FeS, increased φPSI and φPSII were also accompanied by increased abundances of all electron transport complexes[6]. In FeS-OE plants generated here, due to unaltered abundances of other photosynthetic complexes (Fig. 2b), higher φPSI and φPSII clearly demonstrate the effect of the cyt$b_6f$ control on whole leaf electron transport.

In C$_3$ plants, the rate of CEF around PSI increases at high light and generates a higher ΔpH which consequently decreases light harvesting efficiency via generation of a higher NPQ[34]. In line with CEF increasing electron supply to the donor side of PSI[35], no difference in electron flux on the donor side of PSI was detected in plants with higher cyt$b_6f$ abundance at higher irradiance (Fig. 5c). Instead, a higher yield of PSI detected at higher irradiance corresponded to the increased availability of PSI electron acceptors (Fig. 5d). It has been suggested that heme $c_i$ of cyt$b_6f$ exposed to the stroma may be involved in the PGR5-mediated CEF route[36] and cyt$b_6f$, PSI and PGR5 can form a super-complex together with ferredoxin, ferredoxin:NADPH oxidoreductase and PROTON GRADIENT REGULATION5-like1[37]. If this is the case, higher abundance of cyt$b_6f$ could directly increase the rate of PGR5-mediated CEF and contribute to alleviating PSI acceptor side limitation. During the C$_3$–C$_4$ evolutionary transition increased PGR5 abundance appears to have been favoured in both cell types[38] and therefore stimulatory effect of Rieske FeS overexpression on PGR5-mediated CEF could be valid for both mesophyll and BS cells.

C$_4$ plants typically grow in high light environments which allow them to accommodate the higher ATP requirements of the pathway[39]. Higher *pmf* generated by the plants with increased cyt$b_6f$ abundance at high irradiance (Fig. 5e,g), conceivably, indicated a higher capacity for ATP production and could underpin higher assimilation rates (Fig. 4d). Since the effect of cyt$b_6f$ abundance on *pmf* was most prominent above 825 µmol m$^{-2}$ s$^{-1}$, this irradiance might be a prerequisite for a realisation of increased photosynthesis in C$_4$ plants.

Since the electron transport chain in BS cells is specifically optimised for active CEF[27], higher proton-motive force (*pmf*) and ΔpH detected in plants with higher cyt$b_6f$ abundance at high light (Fig. 5e, Fig. 5g) could be present in BS cells. The NDH-mediated route of CEF is considered to be prevalent in bundle sheath cells of NADP-ME plants[40]. Similar protein abundance of the NdhH subunit between FeS-OE and control plants (Fig. 2b) suggested that CEF activity in BS cells could be regulated also at the level of cyt$b_6f$ abundance. Lateral heterogeneity of the thylakoid membranes where PSII is localised to the stacked grana and PSI to the stromal lamellae[41] provides the basis for functional specialisation of cyt$b_6f$. Since about 55% of cyt$b_6f$ is found in appressed grana and 45% is distributed over the stromal lamellae[42], these two fractions of cyt$b_6f$ might be more specific for either linear or CEF. In line with this we detected cyt$b_6f$*complex (Fig. 2c) in BS cells

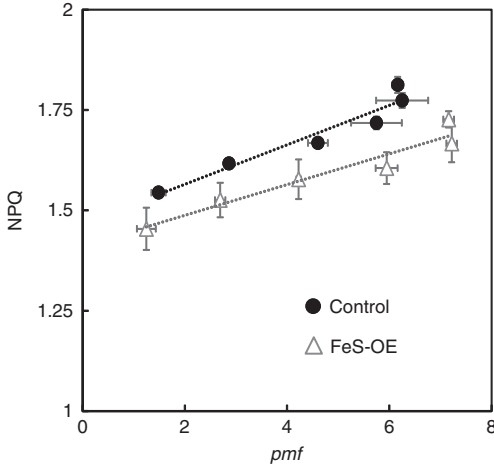

**Fig. 7** The relationship between light-induced proton-motive force (*pmf*) and non-photochemical quenching (NPQ) in plants with Rieske FeS overexpression (FeS-OE) and control plants. NPQ data from Fig. 5a were plotted against the steady-state *pmf* from Fig. 5e

of *S. viridis* that might represent a CEF-specific cyt$b_6f$ potentially forming a complex with its electron transport-partners.

Our results suggest that lower NPQ detected in plants with increased cyt$b_6f$ levels underpinned more efficient use of light energy that could result in higher assimilation rates. Remarkably, higher abundance of Rieske FeS resulted in slower NPQ induction in both $C_3$[6] and $C_4$ plants (Fig. 6) showing an apparent effect on pH-dependent $q_E$ component of NPQ. Moreover, we demonstrate that FeS-OE plants generated the same or higher levels of *pmf* (and $\Delta$pH) compared to control plants despite the decreased NPQ (Fig. 5e). Plants with higher cyt$b_6f$ levels retained linear relationships between *pmf* and NPQ, but showed an offset in the NPQ response (Fig. 7).

Regulation of ATP synthase is thought to play a key role in modulating *pmf* and NPQ[43]. However the effect of Rieske FeS overexpression presented on Fig. 7 was not due to changes in thylakoid proton conductivity (Fig. 5h). In addition, no decrease in violaxanthin content could be detected in *A.thaliana* plants overexpressing Rieske FeS[6] and no difference in PsbS abundance was found between FeS-OE and cotrol plants in this work (Fig. 2b). Although detailed characterisation of the cell-specific changes in FeS-OE plants is required to clarify whether there are alterations in ratios of protein abundance and/or activity of the regulatory mechanisms between mesophyll and BS cells, our observations suggest that there might be other mechanisms regulating NPQ in FeS-OE plants. PGR5-mediated CEF might serve also for redox poising of the electron transport chain[44] to prevent the reduction of $Q_A$ and the closure of PSII centres. If higher cyt$b_6f$ abundance stimulated this pathway, it could result in a sustained change in the redox state of the $Q_A$(i.e. constantly 10% more oxidised $Q_A$) and hence a corresponding offset of the NPQ response in FeS-OE plants (Fig. 7). While lower NPQ might be detrimental for $C_3$ plants in high light conditions, due to higher availability of $CO_2$ as terminal electron acceptor, $C_4$ photosynthesis is more resistant to photo-damage by high irradiance[8] and the "release" of cyt$b_6f$ control could have positive effects on photosynthesis without photoinhibitory effects on photosystems.

At low intercellular $CO_2$ ($C_i$), the assimilation rate is largely limited by the rate of primary $CO_2$ fixation in mesophyll cells[45], whilst at high $C_i$, according to the $C_4$ photosynthesis model[46,47], assimilation rate is co-limited by the amount and activity of Rubisco, the capacity for regeneration of either PEP or ribulose bisphosphate and the rate of whole leaf electron transport (the

sum of electron transport rates of the two cell types). In line with the model, saturating rates of $CO_2$ assimilation were about 8% higher in plants with about 10% higher Rieske FeS abundance (and hence with higher electron transport rate as discussed above). When measured in the steady-state, there was a strong positive correlation between Rieske FeS abundance and $CO_2$ assimilation rate at ambient $C_i$ in three independent transgenic lines (Fig. 3).

While very little work has been done on improving $C_4$ photosynthesis by genetic modification, maize plants overexpressing both subunits of Rubisco together with the chaperonin RUBISCO ASSEMBLY FACTOR 1 demonstrated higher assimilation rates at high $C_i$ and high light[48]. Importantly, since plants with higher cyt$b_6f$ abundance examined here did not have higher Rubisco content, both abundance of Rubisco and cyt$b_6f$ are likely co-limiting factors of $C_4$ photosynthesis under high light and non-limiting $CO_2$, in accordance with the $C_4$ model predictions.

Although increasing electron transport by means of Rieske FeS overexpression is a viable approach for improving $C_4$ photosynthesis, our results also demonstrate that Rieske FeS abundance is tightly regulated on a post-transcriptional level since no increase of protein abundance was detected in homozygous plants with high transcript levels (Fig. 1b, Supplementary Fig. 2). Moreover, overexpression of Rieske FeS in *S. viridis* could only be detected when plants were grown at low irradiance of 200 µmol m$^{-2}$ s$^{-1}$. Due to the long cyt$b_6f$ lifetime of at least one week and fast light-induced degradation of its unassembled components[49], cyt$b_6f$ biogenesis is practically restricted to a short period during the growth of young leaves. Using leaf-specific promoters and overexpressing all cyt$b_6f$ subunits simultaneously might allow these problems to be overcome by ensuring the presence of all parts required for cyt$b_6f$ assembly at the right time.

In conclusion, we provide evidence that cyt$b_6f$ exerts a high level of control over the rate of electron transport in both mesophyll and BS cells in the $C_4$ model plant *S. viridis*. We have demonstrated that increasing cyt$b_6f$ content can help improve light conversion efficiency in both photosystems and support generation of higher proton-motive force providing the basis for higher $CO_2$ assimilation rates. Our results support the idea that $C_4$ photosynthesis at ambient and high $CO_2$ and high irradiance is co-limited by the electron transport rate. Engineering $C_4$ crops with higher cyt$b_6f$ abundance may result in higher $CO_2$ assimilation rates and higher yield.

## Methods

**Construct generation**. Two constructs, 230 and 231, were created using the Golden Gate MoClo Plant Parts Kit[50] (Supplementary Fig. 1). The Golden Gate cloning system allows assembly of multiple independent transcription modules in a plant binary vector pAGM4723. In both constructs the first expression module was occupied by the hygromycin phosphotransferase gene (*hpt*) driven by the rice actin promoter (p*OsAct*). The second construct in both constructs was taken from the coding sequence of *PetC* gene encoding for the Rieske FeS subunit of the Cytochrome $b_6f$ complex from *Brachypodium distachyon* domesticated for the Golden Gate (Supplementary Fig. 1) and driven by the maize ubiquitin promoter (p*ZmUbi*). Construct 231 contained a third expression module with the coding sequence of the cyan fluorescent protein mTurquoise driven by the 2x35S promoter. The bacterial terminator tNos was used in all transcription modules. Both constructs were verified by sequencing and transformed into *Agrobacterium tumefaciens* strain AGL1 to proceed with stable plant transformation.

**Generation and selection of transgenic plants**. Stable agrobacterium-mediated transformation of *S. viridis* (accession A10.1) tissue culture was performed as described in Osborn et al.[51]. Hygromycin-resistant plants were transferred to soil and grown in control environmental chambers with 2% $CO_2$, the irradiance of 200 µmol m$^{-2}$ s$^{-1}$, 16 h photoperiod, 28 °C day, 20 °C night and 60% humidity. $T_0$ plants were analysed for the *hpt* gene copy number by digital droplet PCR (iDNA genetics, UK) as described in Osborn et al[51]. The plants that went through the transformation process and tested negative for the *hpt* gene were used as control for $T_0$ and $T_1$ generations. Zero(Null)-segregates were used as control for $T_2$ plants. $T_1$ and $T_2$ progenies were analysed with digital droplet PCR to confirm the

presence of insertions. MultispeQ v1.0 leaf photosynthesis V1.0 protocol was used for a fast screening of transgenic plants for altered photosynthetic properties[52]. MultispeQ data were analysed with the PhotosynQ web application (https://photosynq.org).

**Plant growth conditions**. The $T_1$ progenies of $T_0$ plants were analysed to select the lines with Rieske FeS overexpression. $T_1$ seeds were sterilised and germinated on medium containing 2.15 g L$^{-1}$ Murashige and Skoog salts, 10 mL L$^{-1}$ 100× Murashige and Skoog vitamins stock, 30 g L$^{-1}$ sucrose, 7 g L$^{-1}$ Phytoblend, 20 mg L$^{-1}$ hygromycin (pH 5.7). Seedlings that were able to develop secondary roots were transferred to 1 L pots with garden soil mix fertilised with 1 g L$^{-1}$ Osmocote (Scotts, Bella Vista, Australia). Plants were grown in controlled environmental chambers under the 16 h photoperiod, 28 °C day, 20 °C night, 60% humidity and ambient $CO_2$ concentrations. Light at the intensity of 200 μmol m$^{-2}$ s$^{-1}$ was supplied by 1000 W red sunrise 3200 K lamps (Sunmaster Growlamps, Solon, OH). Seeds of the $T_1$ plants with confirmed Rieske FeS overexpression were collected to be analysed in the $T_2$ generation. $T_2$ seeds were incubated in 5% liquid smoke (Wrights, B&G foods, Parsippany, NJ) overnight and germinated in 1 L pots with garden soil mix layered on top with 2 cm seed raising mix (Debco, Tyabb, Australia) both containing 1 g L$^{-1}$ Osmocote. Youngest fully expanded leaves of the 3–4 weeks plants were used for all analyses. Plants of different genotypes were placed randomly in growth chambers to reduce any position effects.

**Gas exchange measurements**. Rates of $CO_2$ assimilation were measured over a range of intercellular $CO_2$ partial pressure and irradiances simultaneously with chlorophyll fluorescence using a portable gas-exchange system LI-6800 (LI-COR Biosciences, Lincoln, NE) and a Fluorometer head 6800-01 A (LI-COR Biosciences). Leaves were first equilibrated at 400 ppm $CO_2$ in the reference side, an irradiance of 1500 μmol m$^{-2}$ s$^{-1}$, leaf temperature 28 °C, 60% humidity and flow rate 300 μmol s$^{-1}$. $CO_2$ response curves were conducted under the constant irradiance of 1500 μmol m$^{-2}$ s$^{-1}$ by imposing a stepwise increase of $CO_2$ concentrations from 0 to 1600 ppm at 3 min intervals. Light response curves were measured at constant $CO_2$ partial pressure of 400 ppm in the reference cell under a stepwise increase of irradiance from 0 to 3000 μmol m$^{-2}$ s$^{-1}$ at 2 min intervals. Red-blue actinic light (90%/10%) was used in all measurements.

**Protein isolation and Western blotting**. To isolate proteins from leaves, leaf discs of 0.71 cm$^2$ were collected and frozen immediately in liquid $N_2$. One disc was ground in ice-cold glass homogeniser in 0.5 mL of protein extraction buffer: 100 mM trisaminomethane-HCl, pH 7.8, supplemented with 25 mM NaCl, 20 mM ethylenediaminetetraacetic acid, 2% sodium dodecyl sulfate (w/v), 10 mM dithiothreitol and 2% (v/v) protease inhibitor cocktail (Sigma, St Louis, MO). Protein extracts were incubated at 65 °C for 10 min and then centrifuged at 13,000 g for 1 min at 4 °C to obtain clear supernatant. Protein extracts were supplemented with 4× XT Sample buffer (BioRad, Hercules, CA), loaded on leaf area basis and separated by polyacrylamide gel electrophoresis [Nu-PAGE 4–12% Bis-(2-hydroxyethyl)-amino-tris(hydroxymethyl)-methane (Bis-Tris) gel, Invitrogen, Life Technologies Corporation, Carlsbad, CA] in running buffer (pH 7.3) containing 50 mM 2-(N-morpholino)ethanesulfonic acid, 50 mM trisaminomethane, 0.1% sodium dodecyl sulfate (w/v), 20 mM ethylenediaminetetraacetic acid. Proteins were transferred to a nitrocellulose membrane and probed with antibodies against various photosynthetic proteins purchased from Agrisera (Vännäs, Sweden) in dilution recommended by the producer: PetC (Agrisera, AS08 330), PsaB (Agrisera, AS10 695), D1 (Agrisera, AS10 704), NdhH (Agrisera, AS16 4065), AtpB (Agrisera, AS05 085), Lhca1 (Agrisera, AS01 005), Lhcb2 (Agrisera, AS01 003), PsbS (Agrisera, AS09 533), PEPC (Agrisera, AS09 458), RbcL (Agrisera, AS03 037), Cyt f (Agrisera, AS08 306). Quantification of immunoblots was performed with Image Lab software (Biorad, Hercules, CA).

**Thylakoid isolation and Blue Native gel electrophoresis**. These procedures were done in dim light and at 4 °C (or on ice) to reduce light-induced damage of isolated thylakoid complexes. Ten youngest fully expanded leaves were collected from one plant and midribs were removed. Leaves were cut into 3 mm pieces and ground in 100 mL of ice-cold grinding buffer (50 mM 4-(2-hydroxyethyl)-1-piperazineethanesulfonic acid (Hepes)-NaOH, pH 7.5, 330 mM sorbitol, 5 mM MgCl$_2$) in Omni Mixer (Thermo Fisher Scientific, Tewksbury, MA) at the intensity #10 for 2 s. This fast treatment was used to break only mesophyll cells that don't have suberised cell walls. The homogenate was passed through the 80-μm nylon filter and the filtrate containing mesophyll suspension was collected. All tissues collected on the filter were again homogenised in 100 mL of grinding buffer in Omni Mixer during three 10-s cycles at the intensity #7. The homogenate was passed through a tea strainer and then BS strands from the filtrate were collected on the 80-μm nylon filter. BS strands were further ground in 10 mL of grinding buffer in an ice-cold glass homogeniser.

Mesophyll and BS suspensions were filtered through a layer of Miracloth (Merck Millipore, Burlingtone, MA) and centrifuged at 6000 rpm, 4 °C for 5 min. Pellets were first resuspended in ice-cold shock buffer (50 mM Hepes-NaOH, pH 7.5, 5 mM MgCl$_2$) and centrifuged again. Second time pellets were resuspended in ice-cold storage buffer (50 mM Hepes-NaOH, pH 7.5, 100 mM sorbitol, 10 mM

MgCl$_2$) and centrifuged again. Finally pellets were resuspended in an equal aliquot of the storage buffer, snap-frozen in liquid $N_2$ and stored at −80 °C.

For Blue Native gel electrophoresis aliquots of thylakoid samples containing 10 μg chlorophyll ($a+b$) were taken for solubilisation. Aliquots were centrifuged at 6000 rpm, 4 °C for 5 min and then thylakoids were resuspended in ice-cold sample buffer (25 mM BisTris-HCl, pH 7.0, 20% (w/v) glycerol, 2.5% (v/v) protease inhibitor cocktail) to obtain chlorophyll ($a+b$) concentration of 1 μg μL$^{-1}$. An equal volume of 2% (w/v) of n-dodecyl β-D-maltoside in sample buffer was added to thylakoids and the mixture was incubated for 5 min in darkness at 4 °C. Traces of unsolubilised material were removed by centrifugation at 14,000 × $g$ at 4 °C for 30 min. Supernatants were supplemented with 10% (v/v) of Serve Blue G buffer (100 mM BisTris-HCl, pH 7.0, 0.5 M e-amino-n-caproic acid, 30% (w/v) sucrose, 50 mg mL$^{-1}$ Serva Blue G) and applied to the precast native 3–12% Bis-Tris polyacrylamide gel (Invitrogen, Life Technologies Corporation, Carlsbad, CA). Blue Native gel electrophoresis was run using cathode and anode buffers and voltage regime according to Rantala et al.[53] The gel was scanned, then incubated for 30 min in transfer buffer (25 mM trisaminomethane, 25 mM glycine, 20% methanol, 0.1% sodium dodecyl sulphate) and blotted to a nitrocellulose membrane. Western blotting was then performed as usually.

**Chlorophyll and Rubisco assays**. For chlorophyll analysis, frozen leaf discs were ground using the Qiagen TissueLyser II (Qiagen, Venlo, The Netherlands) and total chlorophyll was extracted in 80% acetone buffered with 25 mM Hepes-KOH. Chlorophyll $a$ and $b$ contents were measured at 750.0 nm, 663.3 nm and 646.6 nm, and calculated according to Porra et al.[54]. Amount of Rubisco active sites was estimated by [$^{14}$C] carboxyarabinitol bisphosphate binding assay as described in Ruuska et al.[22]

**Electron transport and electrochromic shift**. Effective quantum yield of PSII (φPSII)[55] was probed simultaneously with the gas-exchange measurements under red-blue actinic light (90%/10%) using multiphase saturating flash of 8000 μmol m$^{-2}$ s$^{-1}$. NPQ as well as effective yields of photochemical and non-photochemical reactions in PSI were measured with Dual-PAM-100 (Heinz Walz, Effeltrich, Germany) under red actinic light with 300-ms saturating pulses of 10000 μmol m$^{-2}$ s$^{-1}$. The effective quantum yield of PSI (φPSI), the non-photochemical yield of PSI caused by donor side limitation (φND) and the non-photochemical yield of PSI caused by acceptor side limitation of PSI (φNA) were calculated as described earlier[56]. To monitor light-induced response of photosynthetic parameters, leaves were dark-adapted for 30 min to record $F_0$ and $F_M$, the minimal and maximal levels of fluorescence in the dark, respectively. Afterwards saturating pulse was given after pre-illumination with far-red light to record $P_M$, the maximal level of P700 oxidation, and $P_0$, the minimal P700 signal, after the pulse. Next, photosynthetic parameters were monitored by the saturating pulse application every 45 s, first for 8 min under actinic light of 220 μmol m$^{-2}$ s$^{-1}$ and afterwards for 5 min in darkness to record NPQ relaxation. After that photosynthetic parameters were assessed in the same leaf over a range of irradiances from 0 to 1287 μmol m$^{-2}$ s$^{-1}$ at 1 min intervals.

The electrochromic shift signal was monitored as the absorbance change at 515 nm using Dual-PAM-100 equipped with the P515/535 emitter-detector module (Heinz Walz, Effeltrich, Germany). Leaves were first dark adapted for 10 min and the absorbance change induced by a single turnover flash (ECS$_{ST}$) was measured. Dark-interval relaxation of electrochromic shift signal was recorded during 3 min in darkness after 3-min intervals of illumination with actinic light at a stepwise increasing irradiance from 0 to 1287 μmol m$^{-2}$ s$^{-1}$. Total proton-motive force ($pmf$) was estimated from the amplitude of the rapid decay of the electrochromic shift signal and slow relaxation of the signal showed the contribution of proton gradient (ΔpH) and electrochemical gradient (ΔΨ) across the thylakoid membrane[57,58]. $Pmf$ levels and the magnitudes of ΔpH and ΔΨ were normalised against the ECS$_{ST}$. Proton conductivity ($g_H^+$) of the thylakoid membrane through the ATP synthase was calculated as an inverse of the time constant obtained by fitting the first-order electrochromic shift relaxation[59].

**RNA isolation and quantitative real-time PCR (qPCR)**. Frozen leaf discs (0.71 cm$^2$) were ground using the Qiagen TissueLyser II and RNA was extracted using the RNeasy Plant Mini Kit (Qiagen, Venlo, The Netherlands). DNA from the samples was removed using the Ambion TURBO DNA free kit (Thermo Fisher Scientific, Tewksbury, MA), and RNA quantity and quality were determined using a NanoDrop (Thermo Fisher Scientific, Tewksbury, MA). One microgram of RNA was reverse transcribed into cDNA using SuperScript™ III Reverse Transcriptase (Thermo Fisher Scientific, Tewksbury, MA). qPCR and melt curve analysis were performed on a Viia7 Real-time PCR system (Thermo Fisher Scientific, Tewksbury, MA) using the Power SYBR green PCR Master Mix (Thermo Fisher Scientific, Tewksbury, MA) according to the manufacturer's instructions. Primer pairs to distinguish between the PetC gene transcript from S. viridis and B. distachyon were designed using Primer3 in Geneious R9.1.1 (https://www.geneious.com): GCTGGGCAACGACATCAAG and CAAAGGAACTTGTTCTCGGC for SvPetC, GGCTCCGGGAGCAACAC and CAAAGGAACTTGTTCTCGGC for BdPetC. Relative fold change was calculated by the ΔΔCt method, using the geometric mean of the Ct values for three reference genes described in Osborn et al.[51].

**Statistics and reproducibility**. One-way ANOVA was performed for electron transport parameters measured with MultispeQ using the PhotosynQ web application (https://photosynq.org). Measurements collected with MultispeQ at significantly different light intensities were excluded, otherwise no data were excluded from the analysis. For other measurements the relationship between mean values for transgenic and control plants were tested using a two-tailed, heteroscedastic Student's $t$-test (Excel 2016). Eight plants were grown and analysed for each of three transgenic lines in $T_1$ generation. Up to 30 plants were grown for each of three transgenic lines in $T_2$ generation and plants with Rieske FeS overexpression confirmed by immunoblotting comprised 10–20% of plants depending on the line. Three $T_2$ plants with the highest overexpression levels from each line were used for analyses. Experiments were partially replicated in $T_1$ and $T_2$ generations for three independent lines.

**Reporting summary**. Further information on research design is available in the Nature Research Reporting Summary linked to this article.

## Data availability

The datasets analysed in this paper are included in this published article and supplementary information files. MultipeQ data are available at PhotosynQ web application (https://photosynq.org), project ID 3400. Plasmids used for generation of plants with Rieske FeS overexpression can be obtained from Addgene (deposit 77017). Further datasets generated during the current study as well as seeds of the WT and FeS-OE S.viridis are available from the corresponding author on request.

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

## Acknowledgements

We thank Soumi Bala for the help with [14C] carboxyarabinitol bisphosphate binding assays and Tegan Norley for the help with electron transport measurements. This research was supported by the Australian Research Council Centre of Excellence for Translational Photosynthesis (CE140100015).

## Author contributions

S.vC. and R.F. conceived the project, S.vC., R.F., C.R. and M.E. planned experiments, P. L.-C. assembled constructs, M.E. performed experiments, M.E. analysed data and wrote the manuscript with contribution of all authors.

## Additional information

**Competing interests:** The authors declare no competing interests.

