## [Peer Review File · Communications Biology]

Reviewers' comments:

Reviewer #1 (Remarks to the Author):

The manuscript entitled "Overexpression of the Rieske FeS protein of the cytochrome b6f complex increases C4 photosynthesis" reports data obtained with *Setaria viridis* overexpressing a protein that is a subunit of a component of the electron transfer chain. Authors concluded that the overexpression of the Rieske FeS protein increased the amount of cytochrome b6f, which in turn, enhanced CO2 assimilation.

The rationale and objective of the study are clear, and the methods used are appropriated. However, there are a couple of points in the results that need clarification:

- Line 104 ff: It says that T1 of line 230 (4) has higher expression, but no increase in protein level; same for 231(3). Is this only for T1? And about T2? If there is no increase in protein level in these lines, how was the calculation for the 10-15% increased protein level (Fig.1b,c; line 103)? It is not clear.

- Line 110-112: Rieske FeS protein overexpression was detected only under low light conditions (200umol m-2s-1). Did the authors test other light conditions? Which ones? The overexpression is not detected under high light?

Following this and the brief suggestion pointed in the text regarding a post-translational and light-dependent regulation of this protein, what would be the implications for a C4 plant that usually grows under high light? Authors mention that the effect of increased cytb6f abundance is prominent only above 825 umol m-2 s-1 of light (line 256-257). Would this be an artifact due to the growth under low light? Or would be this expected even under high light growth conditions with no overexpression (?) of the Rieske FeS protein?

Minor:

- Line 109: I do not think that the citation of Fig. 2b is correct here. Please check.

- Line 151: Based on the average value it was presented, I would say that the difference between transgenic and Null plants is about 8% and not 10%. Also, add the P-value to show the significance of this difference.

- In addition to what is already written, I suggest including in the conclusions the carbon assimilation increase found in this study since it is a highlight of the title and abstract.

Reviewer #2 (Remarks to the Author):

The manuscript by Ermakova et al. presents the effect of overexpressing the Rieske FeS protein of cytb6f on C4 photosynthesis. In this work, the authors have used *Brachypodium* gene sequence and drove the expression under a maize ubiquitin promoter for overexpression (~110% of control) in *Setaria*. As a result, both mesophyll and bundle sheath cells showed higher accumulation of cytb6f. Effect on C4 photosynthesis was monitored using chlorophyll fluorescence, absorbance and gas exchange measurements.

This study finds that under high light and non-limiting CO2 conditions, higher accumulation of b6f translates into higher CO2 assimilation in a proportional manner (1:1).

As reported by the authors, these findings are not novel in that they have been reported previously for C3 plants. The novelty here lies in the demonstration that it is also the case for C4 plants. This claim is of interest to others in the community and the wider field, especially in light of increasing crop

production to meet global food and bioenergy demand.

The work is convincing and I recommend it for publication but I would suggest a few modifications to strengthen the conclusions, see major comments.

Statistical analyses seem to be appropriate and valid for the claims and the level of detail provided is sufficient to reproduce the work (Line 347: provide light bulb reference; Lines 373 and 406: provide buffer system used and antibodies references and dilutions used; Line 428: is 1 min sufficient to reach a steady-state?).

Major comments:

- Figures 1, 2a, S3: please add a loading control, either another antibody probing AtpB for example or coomassie stain of the membrane. I am concerned with the variation of the TO lines which have no insertion, no PetC transcript but yet a variable level of Rieske (e.g. Fig1a 230.2, 231.1). Is this due to loading issues or are these actual differences?

- Figure 6: it would be nice to have blotted against Rieske here.

- Figure S3: were these samples isolated on the same day of gas exchange measurements or at least within a week? The reason I'm asking is that I've had previous experience with overexpressing lines where the expression varied with age of the same line. That was using the 35S promoter, it might not be as big of a problem with the UBQ promoter. Have you tested variation of expression with age?

- Line 111: what other conditions were tested? If there is a post-translational light-dependent regulation of b6f accumulation, then that might 'endanger' the possibility to improve C4 photosynthesis in high light using this strategy (mentioned line 257). Could you comment?

- Line 120, null is not defined. Do you mean wild type? I recommend changing the name "null" to something else as "null" is usually used to describe non-functional alleles. Null could be "control" and No FeS-OE could be "WT FeS" for wild type level.

- Line 136, this result is beautiful (higher form of b6f in bundle sheath) and merits further investigation! Has it been shown before? I briefly searched for it and found for maize: Majeran et al. 2008 MCP 7 (9) 1609-1638, Figure 4, they do not observe differences in petC distribution between M and BS. See also Hernández-Prieto MA et al. 2019 Physiologia Plantarum. Please comment.

- Line 165: title, add at high light irradiances.

- Line 168: I would be cautious with such statement "in agreement with higher PhiPSII, OE had lower NPQ". There may be cases where both PhiPSII and NPQ are high. The comparison of Figure 4 and 5 would be easier if the same x-axis values were used.

On this same point, lines 213-218: as you mentioned, the increase of PhiPSII could be due to more oxidized QA, and not necessarily be due to more light reaching RC. As you explain in the paragraph starting line 272, it is not evident what regulatory mechanisms are at play here. Investigating this question further might require to analyze changes in a cell specific way. In addition to suggesting that different mechanisms may be at play, it would be nice to discuss the possibility that what you measure is a sum of potentially widely differing processes in BS and M cells (e.g. more PsbS in one cell type, ATPase activity higher in one cell type, etc). For the purpose of the work presented here, line 282, I'd still recommend to check overall zeaxanthin formation in the light (although keeping in mind it might be different between cell-types).

- Line 259: please show data for that statement.

Minor comments:

- Line 52: replace trigs by "triggers"

- Line 61: add effects "on" PSII electron

- Line 68: spell out NADP-"Malic Enzyme" (ME)

- Line 90: could you explain the rationale for choosing Brachypodium sequence?

- Line 98: I suggest moving the sentence line 121 here to justify that per leaf area is an appropriate

comparison.

- Line 121: what does SPAD stands for?
- Line 232: take out between "two" PSI
- Line 245: replace heme x by "heme ci"
- Line 524: replace PhiND by "PhiNA"

We thank reviewers for their comments and provide detailed answers to all comments below. We have highlighted all changes made in the text of the manuscript. We have made changes in Figures 1, 2 and S3 to provide loading control and Figure S5 has been moved from the Supplementary material to the main text as Figure 6.

Reviewers' comments:

Reviewer #1 (Remarks to the Author):

- Line 104 ff: It says that T1 of line 230 (4) has higher expression, but no increase in protein level; same for 231(3). Is this only for T1? And about T2? If there is no increase in protein level in these lines, how was the calculation for the 10-15% increased protein level (Fig.1b,c; line 103)? It is not clear.

In that sentence we wanted to point out the difference between homozygous and heterozygous transgenic plants: homozygous plants of the lines 230(4) and 231(3) showed no increase of Rieske FeS on protein level, whilst heterozygous plants of both lines had increased Rieske FeS protein abundance in both T₁ and T₂ generations. Therefore heterozygous plants of those lines were analysed in all experiments. We have changed the text to clarify this. (LL 107-112).

- Line 110-112: Rieske FeS protein overexpression was detected only under low light conditions (200umol m-2s-1). Did the authors test other light conditions? Which ones? The overexpression is not detected under high light?

We tested a growth light of 500 $\mu\text{mol m}^{-2} \text{s}^{-1}$ but we could not detect an increase of Rieske FeS abundance in those conditions. It has been previously reported that *cytb₆f* has a very long lifetime of at least one week (Hojka M. et al., 2014) and that non-assembled *cytb₆f* components undergo fast light-mediated degradation (Ostersetzer & Adam, 1997). Therefore, we suggest that *cytb₆f* biogenesis is practically restricted to a short period during the growth of young leaves. It is possible that slower growth of plants under lower light is beneficial for Rieske FeS overexpression. This discussion has been added to the manuscript (LL 314-323).

- Following this and the brief suggestion pointed in the text regarding a post-translational and light-dependent regulation of this protein, what would be the implications for a C4 plant that usually grows under high light?

We agree with the reviewer that the approach we used to obtain *Setaria* plants with increased *cytb₆f* abundance might not be successful in C₄ crops grown in field conditions. Indeed, implementing this strategy into C₄ crops might require different construct design, for example, a construct using a leaf-specific promoter or simultaneously overexpressing all *cytb₆f* subunits in the nuclear genome to ensure the presence of all *cytb₆f* components in chloroplasts at the moment of assembly. This discussion has been added to the manuscript (LL 314-323). However we show for the first time that increasing of *cytb₆f* has the desired effect of increasing photosynthetic rate.

Authors mention that the effect of increased *cytb₆f* abundance is prominent only above 825 umol m-2 s-1 of light (line 256-257). Would this be an artifact due to the growth under low light? Or would be this expected even under high light growth conditions with no overexpression (?) of the Rieske FeS protein?

We think that the observed effect of increased *cytb₆f* abundance providing higher *pmf* above 825 $\mu\text{mol m}^{-2} \text{s}^{-1}$ is in line with the C_4 plants' requirements of high irradiance to produce enough ATP and sustain the additional energy demands of C_4 photosynthesis. It is possible however that the exact level of irradiance required for the realisation of increased *cytb₆f* abundance can be shifted in high-light grown plants.

Minor:

- **Line 109: I do not think that the citation of Fig. 2b is correct here. Please check.**

Thank you, this is now corrected to Fig. 1b.

- **Line 151: Based on the average value it was presented, I would say that the difference between transgenic and Null plants is about 8% and not 10%. Also, add the P-value to show the significance of this difference.**

Thank you, this is correct and has now been changed throughout the manuscript. P value has been added (L 161).

- **In addition to what is already written, I suggest including in the conclusions the carbon assimilation increase found in this study since it is a highlight of the title and abstract.**

Thank you, this statement has been added to the conclusions (L 328).

Reviewer #2 (Remarks to the Author):

Line 347: provide light bulb reference;

This information has been added to the description of plant growth condition (LL 364-365).

Lines 373 and 406: provide buffer system used and antibodies references and dilutions used;

Buffer system information has been added to the Materials and Methods (LL 389-392). All antibodies were used in dilution recommended by the producer – this has been added to the materials and methods (LL 394). References to all antibodies used in this study are given in the Reporting summary. For Blue Native gel electrophoresis we used cathode and anode buffers according to Rantala et al. (2018) - this has been clarified in the text (LL 424-425).

Line 428: is 1 min sufficient to reach a steady-state?).

Photosynthetic parameters obtained by “rapid light curves” measurements might not always represent the steady-state of photosynthetic parameters at each irradiance. However, this method is commonly used for a fast estimation of photosynthetic parameters and their dynamics during hanging irradiance.

Major comments:

- **Figures 1, 2a, S3: please add a loading control, either another antibody probing AtpB for example or coomassie stain of the membrane. I am concerned with the variation of the T0 lines which have no insertion, no PetC transcript but yet a variable level of Rieske (e.g. Fig1a 230.2, 231.1). Is this due to loading issues or are these actual differences?**

We agree with the reviewer and we added blots probed with an RbcL antibody to Fig. 1 and Fig. S3 to provide loading control. We used the same protein samples for Fig. 2a and Fig. 6 and therefore these figures are now combined in Fig. 2a to provide loading control. RbcL blotting of T₀ plants also shows high variation of protein abundance on leaf area basis (see new Fig. 1a). This variation might be explained by regeneration of T₀ plants from the tissue culture at different times. These plants, therefore, differ substantially in age and other leaf parameters. This discussion has been added to the manuscript (LL 99-100).

- Figure 6: it would be nice to have blotted against Rieske here.

Figures 2 and 6 have now been combined in a new Fig. 2.

- Figure S3: were these samples isolated on the same day of gas exchange measurements or at least within a week? The reason I'm asking is that I've had previous experience with overexpressing lines where the expression varied with age of the same line. That was using the 35S promoter, it might not be as big of a problem with the UBQ promoter. Have you tested variation of expression with age?

We performed western blotting on those plants 2-5 days before the gas-exchange analysis. Unfortunately we did not test for a variation of the PetC gene expression during the growth of our transgenic plants. However long life time of *cytb_{6f}* of at least one week (Hojka M. et al., 2014) suggests that *cytb_{6f}* biogenesis is practically restricted to a short period during the growth of young leaves and therefore changes of the PetC gene expression during the growth of plants should not play significant role. See discussion LL 314-323.

- Line 111: what other conditions were tested? If there is a post-translational light-dependent regulation of b6f accumulation, then that might 'endanger' the possibility to improve C₄ photosynthesis in high light using this strategy (mentioned line 257). Could you comment?

There is indeed a possibility that our current approach will not be successful in C₄ crop species. However having demonstrated that increasing *cytb_{6f}* does indeed increase photosynthetic rate, we suggest that a different construct design might help overcoming these problems. Please see our response to the Reviewer 1 and discussion LL 314-323.

- Line 120, null is not defined. Do you mean wild type? I recommend changing the name "null" to something else as "null" is usually used to describe non-functional alleles. Null could be "control" and No FeS-OE could be "WT FeS" for wild type level.

As a control for the T₂ progeny we used Null segregants, i.e. plants that harboured the insertion in one allele in previous generation(s) but have lost it due to the gene segregation and are genetically identical to wild type. This is now clarified in the text (LL 116-117). We changed "No FeS-OE" to "Null-level FeS" and added the description to L 122 and Table 1.

- Line 136, this result is beautiful (higher form of b6f in bundle sheath) and merits further investigation! Has it been shown before? I briefly searched for it and found for maize: Majeran et al. 2008 MCP 7 (9) 1609-1638, Figure 4, they do not observe differences in petC distribution

between M and BS. See also Hernández - Prieto MA et al. 2019 Physiologia Plantarum. Please comment.

This is indeed an exciting result and, to our knowledge, it was not demonstrated previously in C_4 plants. Composition of thylakoid complexes on Blue-Native gel can vary depending on multiple factors including plants species, growth conditions and dodecyl-maltoside(DM):Chl ratio. The $cytb_6/f^*$ band that we observed might be specific for *Setaria* grown in our conditions and the 20:1 DM:Chl ratio.

- Line 165: title, add at high light irradiances.

The title has been changed to: "Rieske FeS overexpression plants have lower NPQ and can generate more proton-motive force at high irradiance" (LL 175-176).

- Line 168: I would be cautious with such statement "in agreement with higher PhiPSII, OE had lower NPQ". There may be cases where both PhiPSII and NPQ are high. The comparison of Figure 4 and 5 would be easier if the same x-axis values were used.

We agree with the reviewer and this statement has now been corrected (LL 178-179). Unfortunately we couldn't use the same axis for Fig. 4 and 5 since the maximum actinic light intensity provided by Dual-PAM was $1287 \mu\text{mol m}^{-2} \text{s}^{-1}$. In addition, fluorescence measurements made by Licor and Dual-PAM were not directly comparable because of red actinic light used in Dual-PAM and red/blue actinic light used in Licor.

On this same point, lines 213-218: as you mentioned, the increase of PhiPSII could be due to more oxidized QA, and not necessarily be due to more light reaching RC. As you explain in the paragraph starting line 272, it is not evident what regulatory mechanisms are at play here. Investigating this question further might require to analyze changes in a cell specific way. In addition to suggesting that different mechanisms may be at play, it would be nice to discuss the possibility that what you measure is a sum of potentially widely differing processes in BS and M cells (e.g. more PsbS in one cell type, ATPase activity higher in one cell type, etc). For the purpose of the work presented here, line 282, I'd still recommend to check overall zeaxanthin formation in the light (although keeping in mind it might be different between cell-types).

We agree with the reviewer that the two-cell system of C_4 photosynthesis presents a major challenge for interpretation of the results. Indeed, our next aim is to take a closer look at the cell-specific changes in FeS-OE plants and more detailed characterisation of the NPQ-related machinery, including analysis of the xanthophyll cycle. We hope our future work will help clarify the exact mechanisms of electron transport up-regulation in these plants. We thank the reviewer for this suggestion and have now added this discussion to the text (LL 286-288).

- Line 259: please show data for that statement.

We do not have conclusive data supporting this statement so it has now been deleted (L 261).

Minor comments:

- Line 52: replace trigs by "triggers"

Thank you, this has been corrected (L 52).

- Line 61: add effects "on" PSII electron

Thank you, this has been corrected (L 61).

- Line 68: spell out NADP-"Malic Enzyme" (ME)

This sentence has now been changed (LL 67-70).

- Line 90: could you explain the rationale for choosing Brachypodium sequence?

Our previous attempts of protein overexpression in *Setaria* suggested that introduction of another copy of a gene with the same or similar codon sequence could cause a co-suppression of gene expression (Osborn et al. 2016). To avoid this problem, we have chosen a PetC sequence from *Brachypodium* that is significantly different from *Setaria* but has monocot-specific codons.

- Line 98: I suggest moving the sentence line 121 here to justify that per leaf area is an appropriate comparison.

Unfortunately SPAD and leaf thickness were measured only in T₂ progeny and therefore we cannot use those results as a rationale for T₀ plants. We used leaf area to normalise all parameters for an easy comparison to the gas-exchange data obtained on leaf area basis.

- Line 121: what does SPAD stands for?

SPAD is Soil Plant Analysis Development chlorophyll meter. This has been added to Lines 123-124 and Table 1.

- Line 232: take out between "two" PSI

Thank you, this has been corrected.

- Line 245: replace heme x by "heme ci"

Thank you, this has been corrected (L 248).

- Line 524: replace PhiND by "PhiNA"

Thank you, this has been corrected (L 553).

REVIEWERS' COMMENTS:

Reviewer #1 (Remarks to the Author):

I believe that after the modifications made by the authors, the points were clarified and this manuscript is now suitable for publication.

Reviewer #3 (Remarks to the Author):

The authors have appropriately addressed my comments.

A minor remaining point concerns the usage of the term 'null segregant'. Now I understand that 'zero-segregant' is meant by that. Null and zero may be interchangeable but I would recommend using 'zero' as null is usually used for non-functional or 'null' allele, or better to use 'control' and 'control-level' at least in the figures to optimize reader's comprehension.